# Statins for extension of disability-free survival and primary prevention of cardiovascular events among older people: protocol for a randomised controlled trial in primary care (STAREE trial)

Sophia Zoungas ![ORCID],[1] Andrea Curtis,[1] Simone Spark,[1] Rory Wolfe ![ORCID],[1] John J McNeil,[1] Lawrence Beilin,[2] Trevor T-J Chong,[3,4] Geoffrey Cloud,[4,5] Ingrid Hopper,[1,6] Alissia Kost,[1] Mark Nelson,[7] Stephen J Nicholls,[1,8] Christopher M Reid,[1,9] Joanne Ryan ![ORCID],[1] Andrew Tonkin,[1] Stephanie A Ward,[1,10] Anthony Wierzbicki,[11] On behalf of STAREE investigator group

For numbered affiliations see end of article.

**Correspondence to**
Professor Sophia Zoungas;
sophia.zoungas@monash.edu

## ABSTRACT

**Introduction** The world is undergoing a demographic transition to an older population. Preventive healthcare has reduced the burden of chronic illness at younger ages but there is limited evidence that these advances can improve health at older ages. Statins are one class of drug with the potential to prevent or delay the onset of several causes of incapacity in older age, particularly major cardiovascular disease (CVD). This paper presents the protocol for the STAtins in Reducing Events in the Elderly (STAREE) trial, a randomised double-blind placebo-controlled trial examining the effects of statins in community dwelling older people without CVD, diabetes or dementia.

**Methods and analysis** We will conduct a double-blind, randomised placebo-controlled trial among people aged 70 years and over, recruited through Australian general practice and with no history of clinical CVD, diabetes or dementia. Participants will be randomly assigned to oral atorvastatin (40 mg daily) or matching placebo (1:1 ratio). The co-primary endpoints are disability-free survival defined as survival-free of dementia and persistent physical disability, and major cardiovascular events (cardiovascular death or non-fatal myocardial infarction or stroke). Secondary endpoints are all-cause death, dementia and other cognitive decline, persistent physical disability, fatal and non-fatal myocardial infarction, fatal and non-fatal stroke, heart failure, atrial fibrillation, fatal and non-fatal cancer, all-cause hospitalisation, need for permanent residential care and quality of life. Comparisons between assigned treatment arms will be on an intention-to-treat basis with each of the co-primary endpoints analysed separately in time-to-first-event analyses using Cox proportional hazards regression models.

**Ethics and dissemination** STAREE will address uncertainties about the preventive effects of statins on a range of clinical outcomes important to older people. Institutional ethics approval has been obtained. All research outputs will be disseminated to general practitioner co-investigators and participants, published in peer-reviewed journals and presented at national and international conferences.

**Trial registration number** NCT02099123.

## STRENGTHS AND LIMITATIONS OF THIS STUDY

⇒ STAtins in Reducing Events in the Elderly (STAREE) is a double-blind, placebo-controlled randomised clinical trial among a primary prevention population of people aged 70 years and over that will evaluate the effect of statin therapy on two co-primary outcomes including disability-free survival and major cardiovascular events.

⇒ STAREE is a pragmatic trial run through general practice across metropolitan and regional/rural Australia with usual care provided by general practitioners.

⇒ STAREE is one of the largest statin trials to carry out a comprehensive assessment of several causes of incapacity including dementia and disability.

⇒ Complete vital status ascertainment will be facilitated by linkage to the National Death Index.

⇒ STAREE will be examining the effects of starting statins but not withdrawing statins or de-escalating therapy in older age.

## INTRODUCTION

The world is experiencing a rapid demographic transition towards an older population.[1] Increasing survival among older age groups is a major contributor to this shift, with almost twice as many older people surviving into their 80s and 90s compared with 50 years ago.[2] In the USA, the proportion of the population aged 65 and over is projected to

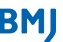

increase from 40 million in 2009 to 89 million in 2050 with 19 million (20%) in the oldest age group (85+) by 2050.[3] Similar trends are seen in other high-income countries such as Australia, Canada, Japan, Europe and the UK.[2] However, over the same time period healthy life expectancy has not increased to the same extent as life expectancy.[4 5] While preventive therapies have reduced the burden of chronic diseases at younger ages, age remains an important determinant of cardiovascular disease (CVD), dementia and frailty.[6–8] The net impact of an ageing population is that more people are living with chronic diseases for longer, and these can lead to disability and loss of independence.[9 10] Loss of independence is not only one of the biggest concerns for older people[11] but has significant societal and economic impacts.[12–14] Identifying preventive interventions that can extend life free of dementia and disability (disability-free survival) is an important public health goal. One promising but insufficiently tested preventive approach to extending disability-free survival among older people is statin therapy.

### The debate about optimal LDL cholesterol levels, use of statin therapy and clinical outcomes in older people

While high levels of LDL cholesterol (LDL-C) are known to increase the risk of CVD events and mortality among young and middle-aged populations,[15 16] the relationship between LDL-C and clinical outcomes including all-cause or cause-specific death in older populations is not consistent. Previous observational studies have reported a mix of no association, positive association or U-curve associations between LDL-C levels and all-cause death,[17–20] cardiovascular events[21 22] and dementia.[16 23] One explanation for the lack of an association or negative association is 'reverse causality', whereby unquantified comorbid disease may be causing both lower LDL-C levels and an increased risk of death or dementia.[24] Given the uncertainty around the relationship of LDL-C levels and clinical outcomes in older people, the decision as to whether lipid-lowering therapy such as statins should be initiated must be guided by randomised controlled trials. Furthermore, absolute CVD risk calculators, whose use is recommended to assist prescribing decisions for primary prevention, are principally driven by age such that universal treatment becomes the default option without certainty of the risks and benefits for older populations.[25]

Statins have the potential to delay the onset of several causes of incapacity in older people. Their best-recognised pharmacological action is to inhibit the mevalonate pathway and biosynthesis of cholesterol and isoprenoids by binding to HMG-CoA reductase. The reduction in circulating lipid levels, especially LDL-C, underpins their ability to reduce the incidence of acute myocardial infarction (MI) and stroke (including subclinical CVD) and their potential long-term sequelae (heart failure, cognitive decline).[26] Statins also exert a number of pleiotropic effects including anti-oxidative, anti-thrombotic and a moderately powerful anti-inflammatory action,[27] which could also help explain their reported beneficial effects in reducing disability and the risk of dementia.[28] The pleiotropic effects of statins have been attributed to the inhibition of isoprenoid metabolite formation (geranylgeranyl pyrophosphate and farnesyl pyrophosphate).[29] These metabolites are intermediates in the post-translational formation of several cell-signalling proteins that control multiple cell functions including maintenance of cell shape, motility, factor secretion, differentiation and proliferation thus explaining improvements in endothelial function, vascular inflammation and oxidation and atherosclerotic plaque stability.[29] Statins also attenuate neuroinflammation, an enduring feature of neurodegenerative diseases through potent suppression of microglia (neuroimmune cells), activation and inhibition of cytokine production, and have been shown to mitigate the aggregation of amyloid β, a key factor in the development of Alzheimer's disease.[30 31]

In people with known CVD, the benefits of statin therapy are clear, with the risks of cardiovascular events reduced across all ages. In people without CVD, the benefits of taking a statin are not as clear at older ages and continue to be debated.[32 33] Most national and international guidelines have not been able to make strong recommendations to guide prescribing for older people, and any recommendations have generally been informed by subgroups within trials that have not studied the range of health outcomes meaningful to older people and that consider both beneficial and adverse effects.[34 35]

A meta-analysis of pooled individual participant data (aged 55+ years) from 28 statin trials that included participants with and without known CVD, reported a 21% reduction in risk of major vascular events per 1 mmol/L reduction in LDL-C, attributed to reduced risks of major coronary events, major stroke events and vascular death.[36] Analyses by age group indicated the greatest risk reductions (25% per 1 mmol/L reduction in LDL-C) among participants aged 55 years or less. This compared with risk reductions of 19% and 13% for older participants aged >70 to ≤75 and >75 years, respectively.[36] The risk of all-cause death was also reduced, although to a much lesser extent (9% per 1 mmol/L reduction in LDL-C). There was no effect on non-vascular deaths including cancer deaths. Of note, participants were predominantly high risk, middle-aged men with high baseline LDL-C levels (mean weighted LDL-C level 3.7 mmol/L) or known CVD.[36] In analyses limited to participants with no history of vascular disease, there was a trend toward smaller proportional risk reductions in major vascular events as age increased.[36] However, a post hoc pooling of subgroup data from two trials of rosuvastatin (the Justification for the Use of Statins in Prevention: an Intervention Trial Evaluating Rosuvastatin (JUPITER) trial, and the Heart Outcomes Prevention Evaluation (HOPE 3) trial, concluded that the effect on major vascular events in people aged ≥70 years (HR 0.74; 95% CI 0.61 to 0.91) was similar to those <65 years (HR 0.75; 95% CI 0.57 to 0.97).[37] Importantly, the effects on dementia and physical

disability were not comprehensively investigated by these studies.[38]

Yet statin treatment is not without potential risk of harm.[39] Observational studies have reported muscle weakness/pain, liver impairment, new-onset diabetes and haemorrhagic stroke[15 40 41] but the incidence of these adverse effects has been much lower in clinical trials.[42] Importantly, statins may interact with other medications, contributing to adverse drug reactions and avoidable hospitalisations.[43] Older adults exposed to polypharmacy are more vulnerable to adverse reactions and increased risks of adverse effects. As with efficacy, the burden of statin adverse effects in older people has not been well studied. As a result, it is still not possible to know the balance of risks and benefits in this age group.[44]

In conclusion, the gains made in increasing lifespan may not have been accompanied by similar gains in healthy life, for the additional later years of life lived. A strong potential exists for greater use of statins for primary prevention to extend healthy life and maintain independence among the ageing population however, current evidence for starting statins among older people is insufficient to support this approach. The STAtins in Reducing Events in the Elderly (STAREE) trial is a large scale randomised controlled trial (RCT) specifically designed to address this need.

### Study objectives and hypotheses
#### Primary objectives
To determine in people aged ≥70 years the effect of statin therapy (40 mg atorvastatin) versus placebo, on two co-primary clinical endpoints:

1. Disability-free survival defined as survival free of dementia and persistent physical disability (a composite of all cause death or development of dementia or persistent physical disability); and
2. Major cardiovascular events (a composite of cardiovascular death, non-fatal MI or non-fatal stroke).

We hypothesise that treatment with a statin (atorvastatin 40 mg) will: (1) prolong overall disability-free survival, and (2) reduce the incidence of major cardiovascular events among a primary prevention population aged ≥70 years.

#### Secondary objectives
To determine the effects of statin therapy (atorvastatin 40 mg) versus placebo on each of all-cause death, dementia and other cognitive decline, persistent physical disability, fatal and non-fatal MI, fatal and non-fatal stroke, cardiovascular death, fatal and non-fatal cancer, heart failure hospitalisation, atrial fibrillation, all-cause hospitalisation, need for permanent residential care and quality of life.

#### Tertiary objectives
To determine the effects of statin therapy (atorvastatin 40 mg) versus placebo on each of fasting glucose/glycated haemoglobin, urine albumin to creatinine ratio (ACR), estimated glomerular filtration rate (eGFR), frailty phenotype and functional independence (instrumental activities of daily living (ADLs)).

## METHODS AND ANALYSIS
### Study design
STAREE is a double-blind, randomised, placebo-controlled primary prevention trial. The WHO Trial Registration Data Set is shown in online supplemental appendices 1 and 2a,b detail the study protocol.

### Setting, participant recruitment, screening, consent and follow-up
Recruitment is being conducted through general practices across metropolitan and regional/rural Australia with participating general practitioners (GPs) registered as STAREE co-investigators. General practice clinical database reviews create a list of potential participants aged ≥70 years without a history of clinical CVD, diabetes, dementia or other life-limiting conditions. Each GP screens the list for suitability and approves the sending of an invitation letter through the mail. Potential participants call the study centre and undergo an initial phone screening for trial inclusion and exclusion criteria. Those who are eligible are invited to take part in two baseline screening visits, either as a face-to-face visit at their usual GP clinic or as a remote phone visit with a trained study team member.

At the first baseline screening visit informed consent is obtained (online supplemental appendix 2a) and then further eligibility assessments (described below) are completed. Demographic information, lifestyle factors, medical history, concomitant medications and other physical measurements (blood pressure, heart rate, height, weight and waist circumference) are collected. Eligible participants then enter a 4-week run-in period and are asked to take one tablet of run-in medication (matching placebo) each day to ascertain medication compliance. Placebo has been chosen for run-in instead of active drug, so participants are not excluded on the basis of statin intolerance and the results are generalisable to the real-world use of statins. Eligible participants also attend their GP clinic for GP review of the trial inclusion and exclusion criteria, as well as any pathology testing.

At the second baseline screening visit after the run-in period, participants who achieve run-in medication compliance of >85% and have GP approval to participate in the trial, complete a cognitive battery to ascertain baseline cognitive performance (see online supplemental appendix 2a).

See online supplemental appendix 2a and b for a detailed description of the schedule of visits and all tests performed.

### Inclusion and exclusion criteria
Individuals are deemed eligible if they; (1) are aged ≥70 years; (2) live independently in the community; (3) are

willing and able to provide informed consent and accept the study requirements and (4) do not have a history of clinical CVD events, diabetes or dementia. Exclusion criteria are a Modified Mini-Mental State Examination (3MS) score <78 on screening, moderate or severe chronic kidney disease, moderate or severe liver disease or serious intercurrent illness likely to cause death within the next 5 years; total cholesterol >7.5 mmol/L; current participation in another interventional clinical trial; and an absolute contraindication to statin therapy, current use of statin therapy or other lipid-lowering therapy for primary prevention and unwilling to stop therapy or current long-term or permanent use of potent cytochrome P450 3A4 inhibitors.

### Randomisation to study treatment

Participants are randomly allocated to either atorvastatin or placebo treatment through a web-based system according to a randomisation schedule generated by an independent statistician. The allocation list was generated using randomly permuted blocks (of size either 2 or 4) with randomisation stratified by state (Victoria, Tasmania, South Australia, Western Australia, Queensland and New South Wales) and age group (70–80 years, ≥80 years). Blocks ensure ongoing approximate balance of randomisation to statin and placebo within strata.

### Study medication

Study medication comprises active treatment with atorvastatin tablets (20 mg) or identical placebo. Participants are asked to take one tablet of study medication (20 mg of atorvastatin or placebo) per day for 4 weeks. They are then contacted by phone to assess study medication tolerability and instructed to increase the dose to two tablets per day (2×20 mg). Dose reduction of study medication to one tablet per day (20 mg) or temporary interruption is permitted throughout trial follow-up to assist in managing tolerability and to reduce study medication discontinuation.

Atorvastatin has been chosen as one of the most commonly prescribed statins in Australia and many other countries with extensive GP experience in its use.[40] Importantly, relative to other statins, it is one of the most potent statins for reducing LDL-C with the dose of 40 mg expected to produce a 30–50% reduction.[45] The 2×20 mg tablet formulation allows for dose adjustments and simplicity of supply of study product.

Medication is posted to participants at approximately 6 monthly intervals. Both active and placebo treatments are supplied by Ramsay Community Pharmacy and dispensed by an unblinded and independent pharmacist through a Monash University-affiliated central pharmacy dispensing centre.

### Study measures and post-randomisation follow-up

The trial activity schedule is summarised in (online supplemental, appendix 2b). Participants are followed via phone or face-to-face visits every 6 months. At follow-up visits, participants are asked about use of concomitant medications including commencement of new medications, adverse events and potential study endpoints. Supporting clinical information relating to any serious adverse events or endpoints is sourced from: (1) medical record review; (2) hospital records and discharge summaries; and (3) linkage with national administrative death, hospitalisation and disease-specific registry data sets where possible.

A random subset of participants will also be offered the opportunity to attend STAREE sites with facilities for processing and storing biospecimens for collection of whole blood and/or urine samples, as outlined in online supplemental appendix 2a and 3.

### Adherence and retention strategies

Adherence to medication is encouraged with multiple strategies including review of study medication use at follow-up phone and face-to-face visits and through reminders included in quarterly participant newsletters and annual face-to-face or web-based study updates.

### Participant well-being

Participants are advised to seek care from their usual treating GP (or other healthcare providers) for any medical condition arising during the study. Any tests or measurements performed as part of the trial trigger a notification to the participant's GP if outside the normal range or deemed clinically significant.

### Commencement of lipid-lowering therapy

Any participants who develop a recognised clinical indication (such as an MI or stroke) for statin treatment can take open label statin as per usual practice by their treating clinician or clinical teams and will cease study medication while continuing in the study for observation (routine follow-up visits).

### Study conduct amendments in response to COVID-19

As a result of government and workplace restrictions implemented during the COVID-19 pandemic, and to ensure safety of staff and participants, amendments to the trial protocol were implemented in 2020 to enable remote screening and follow-up visits by phone instead of in-person visits. After easing of COVID-19 restrictions, participants were given the option of continuing with remote screening and follow-up visits or face-to-face clinic visits depending on their risk profile, personal preference and location.

### Blinding and emergency unblinding

All STAREE staff including the investigator team, management, administration, medical and research officers and students are blinded to treatment allocation through the randomisation procedure and study duration. All GP co-investigators and practice staff will be blinded to treatment allocation. All participants will be blinded to treatment allocation for the study duration. GP co-investigators are requested to not measure lipid levels in

participants after randomisation, to minimise the risk of unmasking.

In the event of a clinical emergency, participants will be assumed to be on active treatment and the recommendation will be to cease study treatment if there was a requirement for statin use. The need for unblinding of study participants will be determined on a case-by-case basis after referral to a senior STAREE medical investigator who will discuss the clinical need for unblinding with the requesting clinician and then authorise the request. The process of emergency unblinding will be overseen by the Monash University-affiliated, independent pharmacist who will have access to the randomisation code.

### Endpoint definitions and adjudication

Ascertainment of primary endpoints will be performed by Endpoint Adjudication Committees of expert clinicians. These Committees will adjudicate events related to possible MI, stroke, dementia, disability, heart failure and trajectory and mode of death. Each subcommittee will have a chair and a minimum of two other members. Two members will conduct adjudications for each event and discordant cases will be resolved via consensus of all committee members.

Disability-free survival is defined as survival free of dementia or physical disability as derived from the endpoints of all-cause death, dementia and persistent physical disability. It is a composite endpoint that has been used in recent clinical trials and enables an evaluation of the net benefit of statin by capturing both beneficial and adverse effects.[35]

Dementia is defined as a significant impairment in cognitive function ($\geq 2$ SD below population norms; online supplemental appendix 2b) that interferes with independence in everyday activities[46] or as a clinical diagnosis based on the medical record. Persistent physical disability is defined as a loss in an activity of daily living (ADL)[47] based on the Life Ability questionnaire at two consecutive visits separated by a period of 6 months. A loss of an ADL (mobility, bathing, transferring, toileting, dressing or feeding) will equate to a self-report of 'a lot of difficulty', 'unable to do this activity' or a requirement for help in performing the activity. If it is not possible to obtain an ADL assessment, notification of assessment for

aged care services will initiate collection of relevant clinical records.

A major cardiovascular event is defined as the first occurrence of cardiovascular death or non-fatal MI or non-fatal stroke. Cardiovascular death is defined as all deaths of cardiovascular causes including fatal MI, fatal stroke, sudden death and death due to other circulatory causes. MI is any ST and non-ST elevation event based on the fourth universal definition of MI.[48] Stroke is any ischaemic or non-ischaemic event defined by International Classification of Diseases, Eleventh Revision (ICD-11) criteria[49] (not including transient ischaemic attack or subarachnoid haemorrhage and classified as disabling or non-disabling using the Modified Rankin Scale). Brain imaging will determine aetiology with ischaemic stroke further delineated according to the Trial of Org 10172 in Acute Stroke Treatment (TOAST) criteria.[50]

### Statistical considerations

The number of participants needed in the trial was originally based on estimated event rates, using the Aspirin in Reducing Events in the Elderly (ASPREE) trial populationDFS[51] which has similar demographic characteristics to the STAREE population, and on the magnitude of the treatment effect anticipated as a result of the LDL-C lowering effect of atorvastatin.

### Modifications triggered by COVID-19

The recruitment rate in Australia was impacted by the COVID-19 pandemic. In addition, STAREE was originally planned as an international endeavour with proposed funding and recruitment across multiple regions including Australia, however in late 2020 it became evident that the proposed funding for recruitment outside of Australia would not be secured. These developments were considered in July 2021 in finalisation of the trial design, reduction in the sample size from an original target of n=18 000 and corresponding power calculation to allow for feasible completion of the trial in Australia alone.

### Sample size

The sample size requirements are summarised in table 1 for detecting relative risk as HRs with a two-sided

**Table 1** Sample size requirement and power justification for the STAREE (STAtins in Reducing Events in the Elderly) trial's co-primary endpoints

| Co-primary endpoint | Effect size (relative reduction in risk for statin group vs placebo in intention-to-treat analysis based on treatment adherence assumptions) | Power | Required number of events | Required number of participants based on assumed event rates and follow-up |
|---|---|---|---|---|
| Major cardiovascular events (first occurrence of cardiovascular death, non-fatal myocardial infarction or non-fatal stroke) | 20.2% | 80.0% | 619 | 9631 |
| Disability-free survival (first occurrence of dementia, persistent physical disability or death) | 14.5% | 83.1% | 1397 | 9631 |

significance level of 5% and adopting a closed testing procedure of the two co-primary endpoints. The key assumptions of the power calculation are:

► Loss to follow-up will occur at a rate of 2% of study participants per annum (loss to follow-up includes those due to death and is expected to be minimised through good retention of participants and the ability to access medical records or perform record linkage for participants who cease in-person assessment). Hence an average of 6.0 years of follow-up time per participant is expected by the anticipated end of the trial in December 2025, at which time the required number of co-primary endpoints are expected to have accrued.

► An age structure of STAREE participants of 56%: 27%: 12%: 5% in the respective age bands 70–74 years: 75–79 years: 80–84 years: 85+ years. These proportions are based on recruitment of the initial approximately 7500 participants to late 2020.

► With treatment adherence statins will reduce the rate of major cardiovascular events by 25% and the rate of disability-free survival by 18%.

► Treatment crossover will occur and impact the HR that will be observed in intention-to-treat analyses. In the first year of follow-up, we assume 8% of participants randomised to statin will cease taking their study medication. Thereafter we assume 3% of participants per annum in the statin group will cease taking study medication. In the first year of follow-up, we assume 1% of participants in the placebo group will commence open-label statin. Thereafter, we assume 3% of participants per annum in the placebo group will commence statin therapy. Further, we assume that there will be no subsequent treatment 're-crossovers'. It is assumed that the effects of statin use and non-use outside use of study medication is the same as the effects of randomised statin and placebo, respectively.

► Rates of the co-primary endpoint death, dementia or persistent physical disability in the placebo group of 18.3 per 1000 person-years (70–74 years), 33.9 per 1000 person-years (75–79 years), 65.7 per 1000 person-years (80–84 years) and 115.4 per 1000 person-years (85+ years) (see online supplemental appendix 4). To allow for a healthy participant effect at recruitment, it is assumed that these event rates will be observed from the start of the third year of an individuals' follow-up and that in the first and second year of follow-up the corresponding rates, respectively, will be zero and half of that of the third year of follow-up.

► Major cardiovascular event rates in the placebo group, according to age at randomisation, of 9.5 per 1000 person-years (70–74 years), 11.5 per 1000 person-years (75–79 years), 18.5 per 1000 person-years (80–84 years) and 24.5 per 1000 person-years (85+ years).

## Statistical analysis

Baseline characteristics of the two treatment groups will be tabulated and imbalance defined as a 0.25 SD difference in means (quantitative characteristics) or an OR of 1.5 (binary characteristics). Given the large sample size, randomisation is anticipated to adequately balance baseline characteristics of participants in the two treatment groups. Hence unadjusted analyses will be considered primary. All primary comparisons between treatment arms will be on an intention-to-treat basis, that is, according to the group to which participants were randomised and without reference to their actual compliance with assigned treatment. Each of the co-primary endpoints will be analysed separately in time-to-event analyses. Event rates (time to first event within each endpoint definition) will be compared between groups using an HR and 95% CI from a Cox proportional hazards regression model fitted to the endpoint, with censoring for individuals not experiencing an endpoint event at their most recent study visit, and a single covariate being an indicator of the group to which the individual was randomised, statin or placebo. The proportional hazards assumption will be tested for each model. Loss to follow-up will be considered a censoring event. This equates to an assumption that data is missing at random given the participant's treatment group and the timing of their loss to follow-up. The adequacy of this assumption will be checked in sensitivity analyses that will include both imputation approaches and adjustment for baseline covariates predictive of propensity for dropout.

A closed testing procedure will be used to allow for the multiple testing arising from two co-primary endpoints. This approach is based on the expectation that cardiovascular benefit will be the main contributor to improved disability-free survival and that a substantial effect of statins on the latter is unlikely in the absence of an effect on the former. First, major cardiovascular events will be tested at alpha=0.05 and, if the major cardiovascular events p value is <0.05 then second, disability-free survival will be tested at alpha=0.05. If the major cardiovascular events p value is not <0.05 than a p value for disability-free survival will not be presented.

Secondary analyses of the co-primary endpoints will be performed using extended Cox proportional hazards regression models based on time(s) to any event within the endpoint definition (for relevant endpoints that can occur more than once). No statistical adjustment will be made for the multiple secondary endpoints and no p values will be presented for the statin versus placebo comparison for these endpoints. The reporting of all secondary endpoint analyses will make clear whether either of the co-primary endpoints were statistically significant.

Subgroup analyses will be undertaken by sex (female, male), age (70–74, ≥75 years), systolic and diastolic blood pressure levels (terciles), body mass index (<25, 25–29.9, 30+ kg/m$^2$), smoking status (never, past and current), 3MS score (above, below median), eGFR (<60, ≥60 mL/min per 1.73 m$^2$) and LDL-C and HDL-C levels (terciles).

The impacts of COVID-19 will be investigated in sensitivity analyses that include examination of participant

event rates by whether their randomisation occurred pre-pandemic, peri-pandemic or post-pandemic, and consider COVID-19 vaccination status (which will be an intercurrent event for those randomised pre-pandemic and a baseline covariate for those randomised post-pandemic), following emerging guidance on analytical approaches and estimand definitions.[52]

### Data and Safety Monitoring Board

An independent committee comprising experts in clinical trials, cardiology and statistics are overseeing data accrual and safety. As outlined in the Data and Safety Monitoring Board (DSMB) Charter (online supplemental appendix 5), the DSMB will monitor participant deaths, serious adverse events, endpoints, data quality and progress of the study. The DSMB will also review any interim analyses on unblinded data in accordance with stopping rules and will provide recommendations to the STAREE Executive Committee.

### Patient and public involvement

STAREE has embedded mechanisms for ongoing participant and public involvement. These include a participant advisory committee, which reviewed study processes and documents, and a formal study that assessed participant preferences about the delivery of informed consent information.[53 54] Presentations to relevant community, social and professional groups have also offered the opportunity for public involvement.

During the trial, participants may provide feedback via access to a free phone line. Participants are also invited to annual trial updates and 'question and answer' sessions. At the completion of the trial, participants will receive a summary of the study results and information about which treatment arm they were allocated to.

### ETHICS AND DISSEMINATION

STAREE is being conducted in accordance with the Declaration of Helsinki (1964, revised in 2008), the NHMRC Guidelines on Human Experimentation, the federal patient privacy (HIPAA) law and ICH-GCP guidelines and the International Conference of Harmonisation Guidelines for Good Clinical Practice. Institutional ethics approval has been obtained from the following:

Monash University Human Research Ethics Committee (Project ID 2787 and 21528).

Curtin University Human Research Ethics Committee (HR113/2015).

RACGP National Research and Evaluation Ethics Committee (14 - 017).

Tasmanian Health and Medical Research Ethics Committee (H0014918).

University of Newcastle Human Research Ethics Committee (H-2016–0266).

The trial is managed and co-ordinated by the STAREE steering committee (appendix 6) which is responsible for developing and approving the protocol and operational plan. The trial central coordinating centre is based at the School of Public Health and Preventive Medicine, Monash University, 553 St Kilda Road, Melbourne 3004.

All research outputs will be disseminated to GP co-investigators and participants, published in peer-reviewed journals and presented at national and international conferences. Authorship of papers will be according to the criteria of the International Committee of Medical Journal Editors and all STAREE publications will list the lead authors and writing group, and the STAREE Investigator group.

On completion of the trial, and after publication of the primary and secondary outcomes of the study, requests for access to de-identified data (to be provided through a secure online environment) may be submitted to the researchers located at the School of Public Health and Preventive Medicine, Monash University, Melbourne, Australia.

### Trial status

Recruitment commenced in November 2015 and was expected to be completed in 2021. However, COVID-19 public health directives and work restrictions resulted in reduced access to general practices across Australia such that recruitment had to be extended for further 18 months.

### DISCUSSION

There is continuing debate on the utility of statins for primary prevention in older people primarily due to the lack of RCT evidence in this population.[33] While no new RCTs of statins have been reported since the conception and commencement of STAREE in 2015 and completion of HOPE 3 in 2016, new analyses of existing RCT data have added to the debate. Age-specific analyses of pooled data from the HOPE 3 and JUPITER RCTs have suggested significant CVD benefits for all age groups above 65 years including a 26% risk reduction in CVD events in individuals aged 70 years and over.[37] A meta-analysis of efficacy and safety of individual participant data from 28 RCTs of statin therapy that included participants over 75 years indicated non-significant 16% and 8% reductions in major vascular events per 1.0 mml/L reduction in LDL-C among 70–75 and >75 year age groups, respectively, without known vascular disease.[36] However, no significant effects on vascular death, non-vascular death, cancer death or all-cause death were observed in either of these older age groups. Guidelines for statin prescription reflect this lack of robust trial evidence, with varying recommendations for apparently healthy older adults over 70 or 75 years, which have led to confusion among clinicians and consumers alike. In contrast, specific data exist for the use of aspirin and anti-hypertensive drugs in this population which has informed guideline committees in making recommendations for intervention.[3 51 55] STAREE will provide new trial evidence to address this critical gap for statins.

STAREE is a large primary prevention trial that is assessing whether starting statins in adults 70 years and older will extend independent and healthy lifespan. While STAREE captures the CVD endpoints that are the traditional primary outcomes of statin trials, we consider that a wider range of outcomes beyond major CVD events is important to evaluate the overall benefit of statins in an older population. To this end, our disability-free survival outcome including detailed ascertainment of dementia using well validated measures of cognition, such as those used in the ASPREE trial[35] and advocated by others,[33] will allow us to determine whether statins reduce major cardiovascular events and prolong survival-free of dementia or physical disability; health outcomes that are highly valued by older people and the general population.

**Author affiliations**
[1]School of Public Health and Preventive Medicine, Monash University, Melbourne, Victoria, Australia
[2]School of Medicine, University of Western Australia, Perth, Western Australia, Australia
[3]Turner Institute for Brain and Mental Health, Monash University, Clayton, Victoria, Australia
[4]Department of Neurology, Alfred Health, Melbourne, Victoria, Australia
[5]Department of Neuroscience, Monash University, Melbourne, Victoria, Australia
[6]Department of Cardiology and General Medicine Unit, Alfred Health, Melbourne, Victoria, Australia
[7]Menzies Research Institute, University of Tasmania, Hobart, Tasmania, Australia
[8]Victorian Heart Institute, Monash University, Clayton, Victoria, Australia
[9]School of Public Health, Curtin University, Bentley, Western Australia, Australia
[10]Centre for Healthy Brain Ageing, School of Psychiatry, University of New South Wales, Sydney, New South Wales, Australia
[11]Metabolic Medicine/Chemical Pathology, Guy's and St Thomas' Hospitals, London, UK

**Contributors** SZ, JJM, RW, LB, GC, TT-JC, MN, SJN, CMR, AT, AW and SAW developed the concept and design of the STAREE trial. SZ, AC, SS and RW developed and wrote the study protocol. SZ, AC, SS and RW wrote the initial drafts of the manuscript. All authors (SZ, AC, SS, RW, JJM, LB, TT-JC, GC, IH, AK, MN, SJN, CMR, JR, AT, SAW, AW) reviewed and revised further drafts of the manuscript and read and approved the final manuscript.

**Funding** STAREE is sponsored by Monash University and has received funding from the National Health and Medical Research Council (NHMRC APP1068146 and APP1161503) and the Heart Foundation of Australia (HF Stroke Prevention grant). STAREE MIND and STAREE HEART have been awarded NHMRC grant funding (#2006611 and #1165440). The funders have had no role in the study design, collection, management, analysis and interpretation of data or in writing or submission of reports or publications.

**Competing interests** SZ has received NHMRC and Australian Heart Foundation research funding as the principal investigator of the STAREE trial; and payment to the institution (Monash University) from Amgen Australia, AstraZeneca, Boehringer-Ingelheim, Eli Lilly Australia, Merck Sharp & Dohme Australia, Novo Nordisk, Sanofi and Servier for consultancy work outside the submitted work. IH has received research funding from NHMRC and Royal Australasian College of Physicians and honoraria for lectures or advisory board participation from Boehringer Ingelheim, Vifor and Eli Lilly. JJM is supported by an NHMRC Leadership Fellowship (IG1173690). MN has served on a Novartis advisory board on lipid management in 2020. CMR is supported through an NHMRC Principal Research Fellowship (APP 1136372). SJN has received research support from AstraZeneca, New Amsterdam Pharma, Amgen, Anthera, Eli Lilly, Esperion, Novartis, Cerenis, The Medicines Company, Resverlogix, InfraReDx, Roche, Sanofi-Regeneron and LipoScience and is a consultant for AstraZeneca, Amarin, Akcea, Eli Lilly, Anthera, Omthera, Merck, Takeda, Resverlogix, Sanofi-Regeneron, CSL Behring, Esperion, Boehringer Ingelheim and Vaxxinity. TT-JC has received honoraria for lectures from Roche. AW has been a clinical trial investigator for Akcea, Amgen, Regeneron and Silence

Therapeutics. He has served as a guideline chair and done consultancy work for the National Institute of Health and Care Excellence in the UK. AT has received research support or honoraria for lectures, advisory board or data monitoring committee participation from Amgen, The Medicines Group, Merck, Novartis and Pfizer. SAW has received payment to attend an advisory meeting on dementia for Roche.

**Patient and public involvement** Patients and/or the public were involved in the design, or conduct, or reporting, or dissemination plans of this research. Refer to the Methods section for further details.

**Patient consent for publication** Not applicable.

**Provenance and peer review** Not commissioned; externally peer reviewed.

**ORCID iDs**
Sophia Zoungas http://orcid.org/0000-0003-2672-0949
Rory Wolfe http://orcid.org/0000-0002-2126-1045
Joanne Ryan http://orcid.org/0000-0002-7039-6325

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
