## [Reviewer comments · BMJ Open]

ARTICLE DETAILS

TITLE (PROVISIONAL)	Statins for extension of disability free survival and primary prevention of cardiovascular events among older people: protocol for a randomised controlled trial in primary care (STAREE trial)
AUTHORS	Zoungas, Sophia; Curtis, Andrea; Spark, Simone; Wolfe, Rory; McNeil, John; Beilin, Lawrence; Chong, Trevor T-J; Cloud, Geoffrey; Hopper, Ingrid; Kost, Alissia; Nelson, Mark; Nicholls, Stephen; Reid, Christopher; Ryan, Joanne; Tonkin, Andrew; Ward, Stephanie; Wierzbicki, Anthony

VERSION 1 – REVIEW

REVIEWER	Strandberg, Timo University of Oulu, Institute of Health Sciences / Geriatrics
REVIEW RETURNED	02-Dec-2022

GENERAL COMMENTS	Overall, I'm very pleased that the STAREE study has been designed and apparently going well despite COVID-19 challenges. This manuscript includes the protocol in detail and a journal-type article. It is well and clearly-written and gives a good overview of this most useful primary prevention trial for ageing societies. It is essential that STAREE includes endpoints important for older people such as disability, dementia and also HRQoL (SF-36). Also STAREE-MIND is an important substudy, because of (unfounded) suspicions of cognitive problems during statin treatment. I don't think I could offer relevant technical comments for the manuscript, as the protocol has been repeatedly revised, but some questions and comments follow, especially for further analyses. - It might be useful to emphasize that STAREE will give information about STARTING a statin for 70+ people. A null or negative result in STAREE would not necessarily mean that statin treatment started before 70 years of age would be futile also in older age (RE: figure in Strandberg TE, et al. Evaluation and treatment of older patients with hypercholesterolemia: a clinical review. JAMA. 2014 Sep 17;312(11):1136-44).- Co-primary endpoints were somewhat differently presented in abstract and Methods, please consider revising.- While it's understandable that deprescribing will not be specifically addressed, some participants will nevertheless discontinue statin, their fate, as well as that of those who were excluded due to high cholesterol (were they ever started a statin in real-life?), would be of interest.- What about other geriatric syndromes, esp. frailty phenotype?- Frailty is a risk factor for disability and would be interesting to know whether statin treatment would prevent frailty. Walk speed, grip strength are probably not measured, but SF-36 items (vitality, physical function), physical activity and weight change would give
--

	possibilities to assess prefrailty and frailty phenotype. (Zaslavsky O, et al. Comparison of the Simplified sWHI and the Standard CHS Frailty Phenotypes for Prediction of Mortality, Incident Falls, and Hip Fractures in Older Women. J Gerontol A Biol Sci Med Sci. 2017 Oct 1;72(10):1394-1400). - While heart failure hospitalization is one of the endpoints, similarly kidney failure could be listed. - It is fine that the word "elderly" has been avoided in the manuscript text. - Finally, I eagerly look forward to the publication of the endpoint results.
--	---

REVIEWER	Marcellaud, Elodie Institute of Neurological Epidemiology and Tropical Neurology
REVIEW RETURNED	10-Jan-2023

GENERAL COMMENTS	Hello, The use of statins in primary prevention is a controversial subject. Currently, it is not known whether statins have an interest in elderly, nor which patients benefit most from them. The endpoints were y chosen to investigate the efficacy and safety of statins in this population. Disability-free survival is very important in the geriatric population. With increasing age, the risk of dependence and institutionalization increases. Maintaining good autonomy in the elderly helps to reduce public health costs. It is interesting to evaluate the impact of preventive therapies in maintaining autonomy. We look forward to the results.
---

VERSION 1 – AUTHOR RESPONSE

Reviewer: 1

Prof. Timo Strandberg, University of Oulu

Comments to the Author:

Overall, I'm very pleased that the STAREE study has been designed and apparently going well despite COVID-19 challenges. This manuscript includes the protocol in detail and a journal-type article. It is well and clearly-written and gives a good overview of this most useful primary prevention trial for ageing societies. It is essential that STAREE includes endpoints important for older people such as disability, dementia and also HRQoL (SF-36). Also STAREE-MIND is an important substudy, because of (unfounded) suspicions of cognitive problems during statin treatment.

I don't think I could offer relevant technical comments for the manuscript, as the protocol has been repeatedly revised, but some questions and comments follow, especially for further analyses.

- It might be useful to emphasize that STAREE will give information about STARTING a statin for 70+ people. A null or negative result in STAREE would not necessarily mean that statin treatment started before 70 years of age would be futile also in older age (RE: figure in Strandberg TE, et al. Evaluation and treatment of older patients with hypercholesterolemia: a clinical review. JAMA. 2014 Sep 17;312(11):1136-44).

Thank you for this suggestion. We have made this point clearer in the summary points and conclusions of the introduction and discussion. See below

Article summary (Page 4)

- STAREE will be examining the effects of starting statins but not withdrawing statins or de-escalating therapy in older age.

Introduction (Page 8)

A strong potential exists for greater use of statins for primary prevention to extend healthy life and maintain independence amongst the ageing population however, current evidence for starting statins among older people is insufficient to support this approach.

Discussion (Page 23)

STAREE is a large primary prevention trial that is assessing whether starting statins in adults 70 years and older will extend independent and healthy lifespan.

- Co-primary endpoints were somewhat differently presented in abstract and Methods, please consider revising.

The descriptions of the co-primary endpoints have been updated so they are the same in the Abstract and Methods.

In Abstract (Page 3)

The co-primary endpoints are disability-free survival defined as survival free of dementia and persistent physical disability and major cardiovascular events (cardiovascular death or non-fatal myocardial infarction or stroke).

In Methods (pages 8 and 9)

Primary objectives

To determine in people aged ≥ 70 years the effect of statin therapy (40 mg atorvastatin) versus placebo, on two co-primary clinical endpoints:

- (i) disability-free survival defined as survival free of dementia and persistent physical disability (a composite of all cause death or development of dementia or persistent physical disability); and
- (ii) major cardiovascular events (a composite of cardiovascular death, non-fatal myocardial infarction or non-fatal stroke).

- While it's understandable that deprescribing will not be specifically addressed, some participants will nevertheless discontinue statin, their fate, as well as that of those who were excluded due to high cholesterol (were they ever started a statin in real-life?), would be of interest.

Questions on any prior regular statin use are included at baseline and will be described in the baseline characteristics of the study population.

- What about other geriatric syndromes, esp. frailty phenotype?

- Frailty is a risk factor for disability and would be interesting to know whether statin treatment would prevent frailty. Walk speed, grip strength are probably not measured, but SF-36 items (vitality, physical function), physical activity and weight change would give possibilities to assess prefrailty and frailty phenotype. (Zaslavsky O, et al. Comparison of the Simplified sWHI and the Standard CHS Frailty Phenotypes for Prediction of Mortality, Incident Falls, and Hip Fractures in Older Women. *J Gerontol A Biol Sci Med Sci*. 2017 Oct 1;72(10):1394-1400).

Other geriatric syndromes are of great interest. Based on the data being collected, two exploratory outcomes that were planned have been added to the protocol paper as suggested as tertiary outcomes i.e. frailty phenotype and instrumental ADLs.

In Study objectives (Page 9-10)

Tertiary objectives

To determine the effects of statin therapy (atorvastatin 40mg) versus placebo on each of fasting glucose/HbA1c, urine ACR, estimated glomerular rate (eGFR), frailty phenotype and functional independence (instrumental ADLs).

The study protocol and appendices) have been updated to include the addition of these tertiary outcomes and the protocol (v 3.1 parts 1 and 2, respectively has been submitted to and approved by the Monash University Human Research Ethics Committee (MUHREC). The amended protocol and approval certificate are included in this revision submission.

- While heart failure hospitalization is one of the endpoints, similarly kidney failure could be listed.

Kidney function and impairment will be assessed through annual measures of eGFR and UACR (tertiary outcomes) as well as participant reported SAEs and AEs.

- It is fine that the word "elderly" has been avoided in the manuscript text.

- Finally, I eagerly look forward to the publication of the endpoint results.

Thank you.

Reviewer: 2

Dr. Elodie Marcellaud, Institute of Neurological Epidemiology and Tropical Neurology

Comments to the Author:

Hello,

The use of statins in primary prevention is a controversial subject. Currently, it is not known whether statins have an interest in elderly, nor which patients benefit most from them.

The endpoints were y chosen to investigate the efficacy and safety of statins in this population.

Disability-free survival is very important in the geriatric population. With increasing age, the risk of dependence and institutionalization increases. Maintaining good autonomy in the elderly helps to reduce public health costs. It is interesting to evaluate the impact of preventive therapies in maintaining autonomy.

We look forward to the results.

Thank you.